# An Ultrasound Prototype for Remote Hand Movement Sensing: The Finger Tapping Case

**DOI:** 10.3390/s25010123

**Published:** 2024-12-28

**Authors:** Stefano Franceschini, Maria Maddalena Autorino, Michele Ambrosanio, Vito Pascazio, Fabio Baselice

**Affiliations:** 1Department of Engineering, University of Napoli Parthenope, Centro Direzionale, 80143 Napoli, Italy; mariamaddalena.autorino.001@studenti.uniparthenope.it (M.M.A.); vito.pascazio@uniparthenope.it (V.P.); fabio.baselice@uniparthenope.it (F.B.); 2Department of Economics, Law, Cybersecurity, and Sports Sciences, University of Napoli Parthenope, Via della Repubblica 32, 80035 Napoli, Italy; michele.ambrosanio@uniparthenope.it

**Keywords:** finger tapping, bradykinesia, ultrasound, short-range remote sensing, Dopper effect, Parkinson disease

## Abstract

In the context of neurodegenerative diseases, finger tapping is a gold-standard test used by clinicians to evaluate the severity of the condition. The finger tapping test involves repetitive tapping between the index finger and thumb. Subjects affected by neurodegenerative diseases, such as Parkinson’s disease, often exhibit symptoms like bradykinesia, rigidity, and tremor. As a result, when these individuals perform the finger tapping task, instability in both the tap rate and finger displacement can be observed. Currently, clinicians assess bradykinesia by visually observing the patient’s finger tapping movements and qualitatively rating their severity. In this work, we present a novel ultrasound contactless system that provides quantitative measurements of finger tapping, including tap rate and finger displacements. The system functions as an ultrasound sonar capable of measuring the Doppler spectrum of waves reflected by the hand. From this spectrum, various characteristics of the hand movement can be extracted through appropriate processing techniques. Specifically, by performing time–frequency analysis and applying specialized data processing, tapping rates and finger displacements can be estimated. The system has been tested in real-world scenarios involving volunteer finger tapping sessions, demonstrating its potential for accurately measuring both tap rates and displacements.

## 1. Introduction

Many diseases affecting the central nervous system, for example, Parkinson’s and Alzheimer’s diseases, autism, and cerebral palsy, impact the motor functions of the subjects [1,2,3]. For this reason, motor performance evaluation can be helpful in the diagnosis and follow up [4,5]. Focusing on hand movement, typical symptoms of neurodegenerative pathologies are upper limb rigidity, tremors, and bradykinesia [6]. In addition to visual inspection by experts, recently, several technological solutions have been proposed in order to acquire information, which can be divided into contact and contactless approaches. In the first category, accelerometers, gyroscopes, or electromyographic sensors are exploited, often nested into gloves [7,8,9]. Such approaches are able to measure movements with high accuracy, but the use of gloves may interfere with hand movement. On the other hand, contactless systems, which often use optical and depth cameras [10,11], do not have this limitation. Within this category, we can also include radar- and sonar-based approaches [12,13,14,15,16,17,18]. Compared with contact systems, contactless systems have a lower accuracy. In this manuscript, an ultrasound system able to provide quantitative information about hands movements is presented. Despite being a contactless approach, the prototype presented here is able to acquire accurate information, as well as being cheap and easily transportable.

A finger-tapping test case was chosen to evaluate the performances of the proposed prototype. The finger-tapping movement involves repeatedly tapping the index finger and thumb together. Generally, participants are asked to execute the taps with a large finger displacement and rate. Currently, finger-tapping is a clinical test commonly included in the routine diagnostic assessment for Parkinson’s disease [19].

Recently, several studies regarding finger-tapping detection and analysis have been published. For example, techologies include accelerometers [20], smartphone’s touch screen [21], special gloves equipped with inertial sensors [22], leap motion devices [23], magnetic sensors [24], and fiber optics [25]. Finger-tapping has also been monitored in a contactless way. Specifically, the authors of [26] proposed finger-tapping estimation based on RGB and depth cameras. The tap rates were measured using a smartphone microphone in [27]. A kinect system was considered for finger-tapping monitoring in [28]. Finally, in [29], a C-band radar was exploited for finger-tapping sensing. In this communication, we propose a contactless solution based on ultrasound sensors. In contrast to contact systems, the proposed solution does not affect gesture execution. Furthermore, it is worth underlining that most of the contactless systems proposed in the literature are mainly focused on the estimation of the tap rate. Our solution is designed to estimate the instantaneous velocity and displacement of the finger, in addition to the tap rate. Additionally, our system presents better resolution than camera-based solutions [26], and, compared to radar-based solutions [29], this ultrasound-based one is lighter and cheaper. The proposed system was tested with the help of volunteers simulating different tapping scenarios. The results are encouraging for both the tap rate and finger displacement measurement.

In Section 1, the proposed ultrasound system is described, giving details about both the adopted hardware solution and the processing algorithms. Section 3 describes the acquisition protocol and the test conducted for validating the system, and the related results are presented and discussed. Finally, some conclusions are reported.

## 2. Proposed Acquisition System

In this section, the ultrasound finger-tapping monitoring system is presented. The first subsection provides a brief discussion of the basic principle of the system with a focus on the hardware component. Subsequently, both the tap rate and finger-displacement algorithms are described.

### 2.1. Prototype Overview

The developed prototype is shown in Figure 1. Two ultrasound sensors are implemented, a transmitter and a receiver (respectively 40LT16 and 40LR16, manufactured by SensComp). The transmitted signal, a 40 kHz monochromatic sine curve, is generated by a voltage controlled oscillator. The pressure wave scattered by the hand is acquired and processed by ah ad-hoc electronic circuit, designed in order to extract the amplitude and the phase of the base-band demodulated signal. It is worth to note that the After the interaction with the subject’s hand the echo wave is acquired. Considering a moving target the frequency of such echo pressure wave is modified according to the size, the velocity and the orientation of the target following the principles of the Doppler effect. In the specific scenario of finger tapping, the main target is a moving hand and, therefore, analyzing the ultrasound received wave it is possible to obtain quantitative information about the finger’s movements in the radial direction. For this purpose, ad-hoc electronics has been designed in order to extract the baseband signal, where the Doppler information is concentrated, through a demodulation process. More detail on this step can be found in [30]. The transmitted signal is 40 kHz sine wave resulting in, approximately, 8.5 mm of wavelength in air. Such frequency is well suited for the evaluation of the tiny movements as demonstrated in [18,30,31]. Such signal, is acquired by a analog-to-digital converter (mod. USB-6343, National Instruments, Austin, TX, USA) with a sample frequency of 5 kHz and a sixteen-bit precision. Once acquired, the base-band signal is processed by a personal computer in Matlab environment. For the sake of clarity, a block diagram of the proposed system is shown in the bottom of Figure 1.

### 2.2. Signal Acquisition and Analysis

The location and orientation of the US sensors had a clear impact on the measurements. In our acquisitions, the sensor board was placed to form an angle of about 60° from the plane on which the fingers moved. This choice was a trade-off between maximizing the cross-section of the target and the projection of the finger velocity in the radial direction. A picture of the system during the finger-tapping measurement is shown in Figure 2a. With this configuration, the contribution of the index finger is maximized while the contribution of the thumb is almost suppressed. Therefore, the behavior of the acquired baseband signal follows basically the movements of the index finger. In more detail, this signal is a sine wave whose frequency and amplitude change over time. Therefore, it is non-stationary and, consequently, both frequency and time information are fundamental for its pricessing. For this reason, the first step of the processing chain is a computation of the time/frequency transformation of the demodulated signal. Among all the possible solutions, the proposed system used the Short-Time Fourier Transform (STFT). The absolute value of this algorithm is the so-called spectrogram, a two-dimensional matrix containing the combined time–frequency information. An example of the spectrogram (in dB) of a demodulated signal related to a finger-tapping movement in the aforementioned set up is shown in Figure 2b.

Based on observations related to this representation, we designed the algorithms described in the following subsections. More specifically, the biggest contributions are represented by the yellow and light blue color in the image. In the case of a finger-tapping measurement, a single tap is represented by a sequence of two vertical lines corresponding to the opening and closing of the fingers. In particular, the configuration of Figure 2a was designed to track the movements of the index finger, while the vertical light blue lines in Figure 2b correspond to the movement of this finger during tapping.

The first consideration is related to the maximum Doppler shift. Based on our acquisition campaign, during finger-tapping, the doppler components related to the fingers movements do not exceed the range of [−500, 500] Hz, which correspond a range of about [−4.3, 4.3] m/s in velocity.

The second consideration is related to the sign of the frequencies. In the current set up, regarding the upper half of the spectrogram, i.e., negative frequencies, the index is moving away from the sensor board; therefore, all of the related lines refer to the opening phase of the tapping gesture. The other half, conversely, refers to the index approaching to the board, i.e., the closing part of the gesture.

### 2.3. Tap Rate Evaluation Algorithm

Our first goal is to identify the tapping movement from the acquisition. For this purpose, the starting point is the time/frequency representation of the demodulated signal (STFT), as shown in Figure 3a. From this, the power in the band [50, 500] Hz is computed, and a mono-dimensional signal P(t) is obtained:(1)$$P\left(t\right)=\int_{50}^{500}{\left|X\left(t,f\right)\right|}^{2}df,$$
where X(t,f) is the STFT of the baseband signal. In Figure 3b, the normalized instantaneous power after a filtering operation step which is helpful for removing the residual thumb Doppler contribution. The tap time instant is identified by applying a peak detection algorithm (actually it is the instant of the maximum radial velocity in the closing phase). By computing the reciprocal of the delay between consecutive taps it is possible to retrieve the tap rate, as shown in Figure 3c.

### 2.4. Finger-Displacement Evaluation Algorithm

For the finger-displacement computation, the first step is to retrieve the finger velocity over time. For the tap rate algorithm, once again, the starting point is the spectrogram. Considering the same acquisition scenario as in Figure 2a, the Doppler shift $f_{D}\left(t\right)$ is linearly related to the radial velocity according to the following equation:(2)$$v\left(t\right)=\frac{c}{f_{0}}f_{D}\left(t\right),$$
where *c* is the speed of sound in the air (we assumed 343 [m/s]) and $f_{0}$ is the frequency of the transmitted wave (in our case 40 kHz). Considering that, during a tap movement, the index produces the maximum Doppler shift, the finger radial velocity $v_{f}\left(t\right)$ is the following:(3)$$v_{f}\left(t\right)=\frac{c}{f_{0}}f_{D,max}\left(t\right),$$
where $f_{D,max}\left(t\right)$ is the maximum frequency component within the Doppler spectrum.

The results of Equation (Equation 3) are shown in Figure 4b. Given the finger radial velocity, the $i_{th}$ tap index displacement $D_{i}$ can be computed by integrating such a function over the tap time window [$T_{i}$, $T_{i}+\Delta T$]:(4)$$D_{i}=\int_{T_{i}}^{T_{i}+\Delta T}v_{f}\left(t\right)dt,$$

In Figure 4c, the index displacements over time for the considered example is shown.

## 3. Results and Discussion

In this section, the performance of the proposed system is evaluated in several scenarios involving a single healthy volunteer. In more detail, the accuracies of both the tap rate and finger displacement estimation algorithms are measured. For each acquisition, the volunteer was asked to place their hand approximately 30 cm away from the US transducers, as shown in Figure 2a. Each acquisition lasted 7 s and the adopted sampling frequency was 5 kHz.

### 3.1. Tap Rate Estimation

For the tap rate estimation task, three different acquisitions were carried out. In the first scenario, the subject was asked to execute the finger-tapping regularly at a high speed. In the second scenario, they were asked to adopt a lower tap rhythm, but preserving the regularity. In the last scenario, the subject started with a fast tap rate and progressively reduced it.

The spectrograms related to the three acquisitions are shown in Figure 5, while the tap rate estimation algorithm results are reported in Figure 6. From the figures, it is clear that the accuracy of the proposed system allows for the discrimination of the different scenarios. In addition, the system is effective at retrieving the different tap rates and evaluating the regularity of the tap timing over time.

In Figure 7, the boxplots related to the three scenarios are reported in order to highlight the effectiveness of the proposed system. In more detail, with this representation, both the median value of the tap rate of the subject (red lines) and its regularity over time (size of the boxplot) can easily be evaluated. Furthermore, as a synthetic metric, we also evaluated the variance of the instantaneous tap rate. While the first two scenarios had a variance of 0.01 ad 0.09 s, respectively, the third was 0.33 s, clearly indicating its non-regular behavior.

### 3.2. Finger Distance Estimation

The performances of the index finger-displacement evaluation algorithm were analyzed in three different scenarios, as in the previous analysis. In the first one, the subject was asked to execute the finger-tapping regularly, with a fixed finger aperture. In the second scenario, the test was repeated with a smaller finger displacement and preserving the regularity of the movement. In the last scenario, the subject started with a wide finger displacement and reduced it progressively. The spectrograms related to these three scenarios are shown in Figure 8. The results of the finger-displacement algorithm in terms of the instantaneous tap rate are shown in Figure 9. From both figures, it is possible to note that the stability of scenario 1 and 2 were correctly represented by the related displacements. In addition, it is also possible to note the sensibility of the prototype in quantitatively distinguishing the two different finger distances. Regarding the last scenario, represented by the green line in the figure, it is possible to note the progressively decrement in the displacement, demonstrating the capacity of the system to correctly follow this irregularity in tapping.

For further analysis, the boxplots related to all three scenarios are shown in Figure 10. The vertical size of each boxplot can be considered an indicator of the displacement stability. Furthermore, the variance of the displacements was computed. While scenario 1 and 2 presented a variance of 0.01 cm, the variance of scenario 3 was 0.18 cm, showing its non-regular behavior.

Based on the reported results, the system shows promising discrimination accuracy between stable and unstable tapping, both in terms of rates and displacement. In particular, statistical values (e.g., variance) can be used as quantitative metrics for the execution of normal finger tapping and, therefore, establish thresholds for discriminating between healthy and pathological persons.

## 4. Conclusions

In this paper, a novel system designed for the quantitative analysis of hand movements is presented. Akin to an ultrasound sonar, the prototype extracts the Doppler components from the backscattered signal, which allows precise movement measurement. The system’s performance has been evaluated in the case of a finger-tap gesture. By means of a Short-Time Fourier Transform and ad hoc-developed algorithms, it is possible to measure the instantaneous tap rate and finger displacement. In this feasibility study, the prototype is tested in different scenarios with simulated tapping with different speeds and finger displacements in a regular and non-regular fashion, and a good sensibility of the system is found. The prototype is compact and cheap, and achieves promising results in being able to discriminate between stable and unstable scenarios. Such results demonstrate the potential of the system for clinician support in neurodegenerative tests. In particular, with defined statistical characteristics of physiological control subjects’ finger taps (both in terms of rates and displacements), it is possible to define thresholds of “normal” executions and, therefore, quantitatively identify possible pathological scenarios. In the future, we plan to test the proposed system in real pathological scenarios, measuring the discrimination accuracy between controls and pathological subjects.

## Figures and Tables

**Figure 1 sensors-25-00123-f001:**
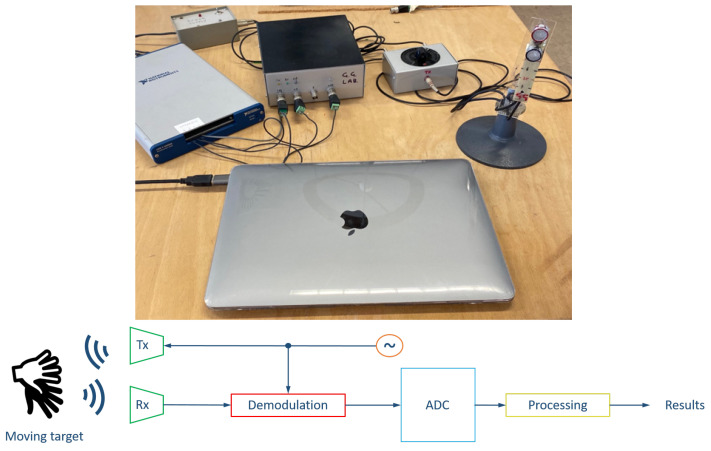
In the top figure, a picture of the ultrasound finger tapping sensing prototype is shown. The acquisition board is on the top left, electronics for signal generation and demodulation are shown in the top center, and the US sensors are on the top right. In the bottom figure, the block diagram of the proposed system is shown. A 40 kHz oscillator pilots the ultrasound transmitter. The generated wave, after interaction with the moving hand, is reflected and sensed by the receiver. The Doppler component of the signal is extracted by the demodulation block, acquired by an acquisition board and, subsequently, processed by a PC.

**Figure 2 sensors-25-00123-f002:**
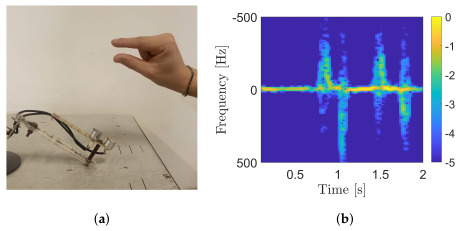
(**a**) Acquisition set up. (**b**) Example of a spectrogram (in dB) related to a finger-tapping gesture.

**Figure 3 sensors-25-00123-f003:**
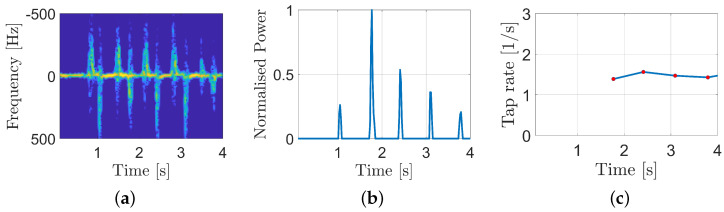
Peak detection algorithm in three steps. (**a**) The spectrogram of a finger-tapping measurements. (**b**) The instantaneous normalized power in the band [50, 500] Hz. (**c**) The reciprocal of the inter-peak delay, i.e., the tap rate.

**Figure 4 sensors-25-00123-f004:**
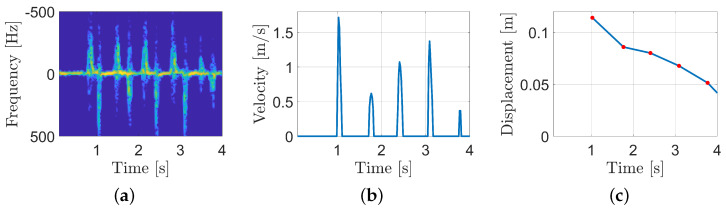
Finger displacement measurement algorithm in three steps. (**a**) The spectrogram of finger-tapping measurements. (**b**) The instantaneous velocity of the fingers. (**c**) The finger displacement related to each tap.

**Figure 5 sensors-25-00123-f005:**
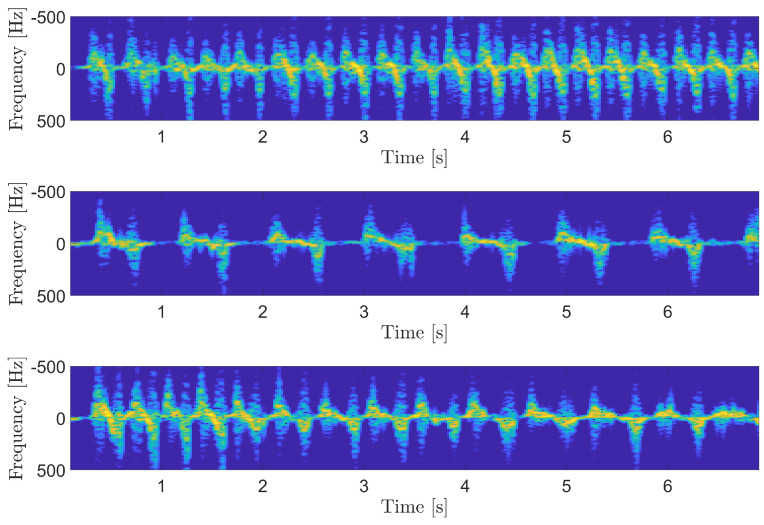
Spectrograms of the three considered scenarios: fast regular (**top**), slow regular (**central**), and decreasing (**bottom**) tapping rate.

**Figure 6 sensors-25-00123-f006:**
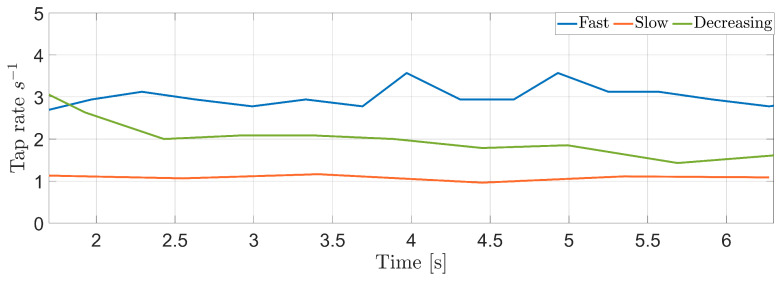
Instantaneous tap rates measured in the considered scenarios: fast regular (blue line), slow regular (orange line), and decreasing (green line) tapping.

**Figure 7 sensors-25-00123-f007:**
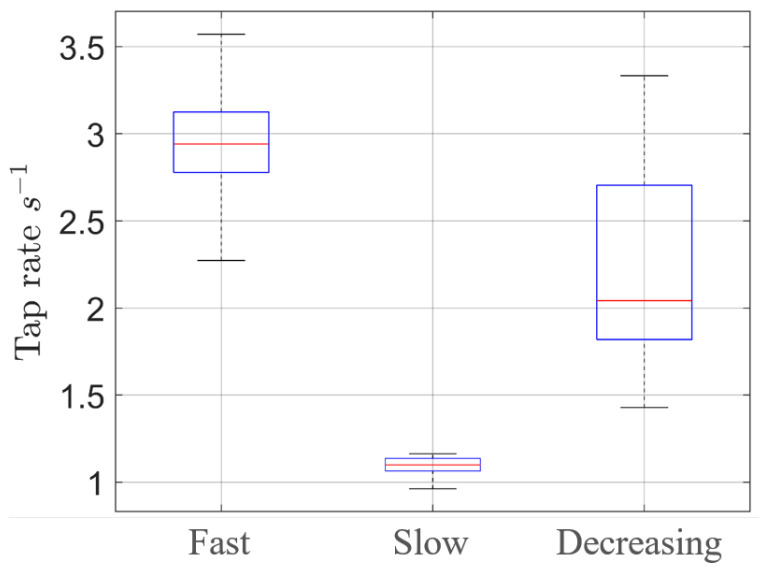
Tap rate boxplots. From the figure, it is possible to note that, the mean tap rate for the fast case is almost the triple of the slow case, while the size is quite similar. The decreasing case, conversely, presents a bigger box, indicating the greater variance in the instantaneous rate.

**Figure 8 sensors-25-00123-f008:**
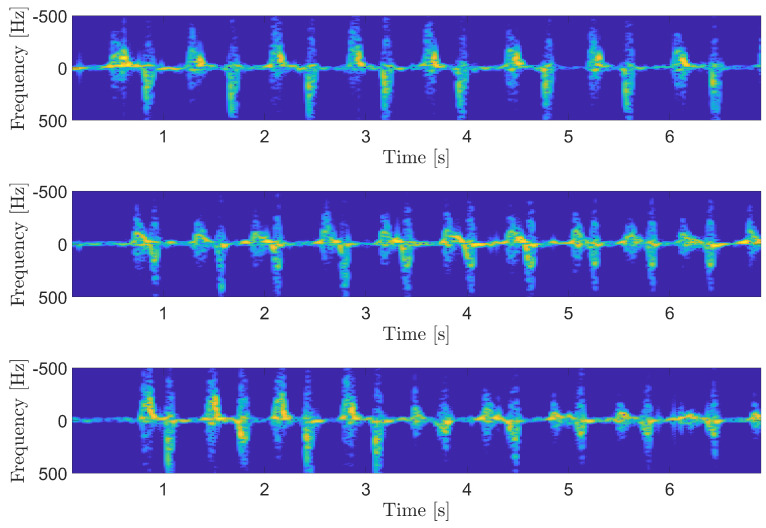
Spectrograms of the three scenarios for finger-displacement testing. On **top**, wide regular tapping is shown. The **center** shows the narrow regular tapping. In the **bottom**, the decreasing displacement is represented.

**Figure 9 sensors-25-00123-f009:**
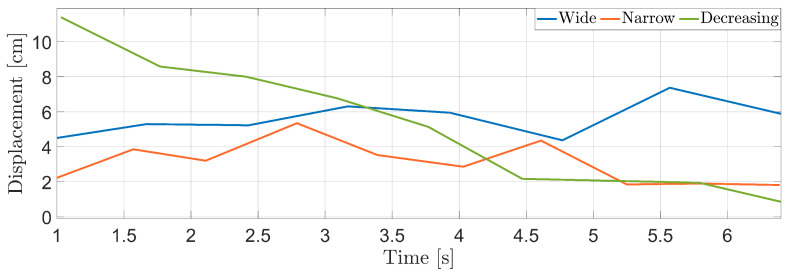
Instantaneous finger displacements. The blue line is the wide scenario and the orange line is related to the tiny displacement scenario. The green line, finally, shows the decreasing displacement tapping.

**Figure 10 sensors-25-00123-f010:**
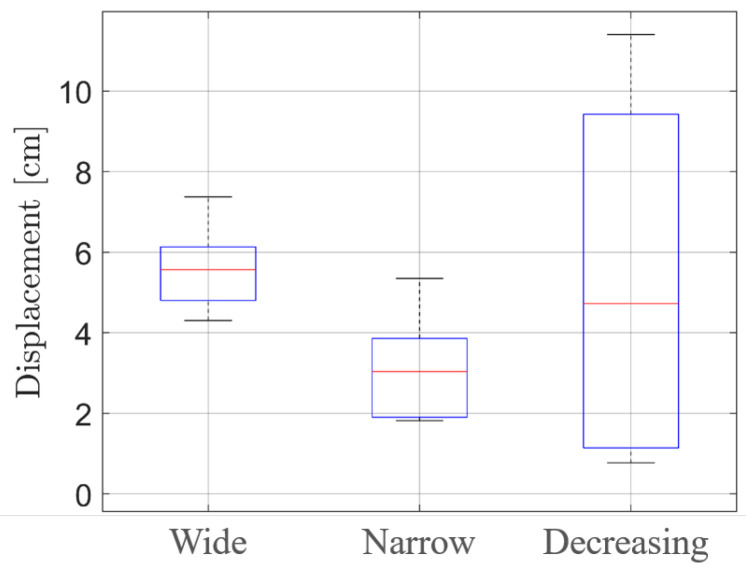
Finger-distance boxplots. From the figure, it is possible to note that the mean displacement for the wide case is almost double that of the narrow case, while the box size is quite similar. The decreasing case, conversely, presents a bigger box, indicating the bigger variance of the displacements.

## Data Availability

All the data are available on request to the authors.

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
