# Peer review of "An Ultrasound Prototype for Remote Hand Movement Sensing: The Finger Tapping Case"

_sensors, 2024, doi:10.3390/s25010123_

Round 1

Reviewer 1 Report

Comments and Suggestions for Authors

Authors proposed remote hand movement sensing using ultrasound technique. However, the analysis of the results are only very limited.  In addition, some results are not reliable. Therefore, I must say the submitted manuscript rejected and then, authors need to do more experiments and analysis of the proposed research with enough time.

1. Please correct Fig. to Figure.

2. In Reference section, authors need to provide city, country, and date information for conference papers.

3. Reference format need to be corrected according to MDPI author guidelines such as abbreviated journal names.

4. There are no analysis with noise and harmonics in spectrograms when displacement is existed.

5. Please correct KHz to kHz. 

6. In Figure 4c, there is no data of Tape rate in 1 s but there is displacement in 1 s in Figure 5c.

7. Authors mentioned some Equations but some information of the label are undefined so please check that in entire manuscript.

8. Please correct 5kHz to 5 kHz.

9. In Conclusion section, authors must provid in detail information of the analysis of the results and limitation. 

10. In Figure 11, there are too broad ranges for Decreasing compared to Wide and Narrow. If this happens, the experiment results seems to be unreliable. Is there any way to decrease the displacement ? If it is not possible, authors must explain the reason.

11. In Figure 3(a), authors had better show entire acquistion setup as shown in Figure 2.

12, In Introduction, authors must provide advantages and disadvantages of the previous study in detail because the analysis of the previous study is too simple.

Comments on the Quality of English Language

English grammar looks uncorrected due to several typos and some sentences are not smoothly described so some English corrections are needed.

Reviewer 2 Report

Comments and Suggestions for Authors

This manuscript was well writen. The opinion of the article was novelty and the content was meaningful. The methods of the study were scientific and reasonable. However, I have two suggestions to further improve the quality of the manuscript.  

1. Figure 1 and Figure 3A should be modified to diagrammatic sketch but not photgraph.The author cpuld provide an additional video that can showcase the entire experimental process.

2. Althogh the quality of English does not limit my understanding of the research, the language of the manuscript could be further improved.

Reviewer 3 Report

Comments and Suggestions for Authors

Dear Erudite Editors of Sensors and Esteemed Authors of the manuscript sensors-3315504, thank you for the invitation to review this interesting work. The manuscript is well-written, and the methodology lacks serious flaws. The results are well-presented and do not give doubts. However, before the Editor can decide, I have a few comments or suggestions for the Authors.

1.      The abstract must be organized into imaginative headings. The way it is written now is meaningless. Please rewrite the abstract following the headings introduction, methods, results, and conclusions.

2.      The discussion section must be added to this manuscript. In the introduction, several previous studies are mentioned. The Authors must retrieve these studies and compare them to the present study to assess similarities and differences.

3.      The conclusion section must be completed with the possible clinical implications derived from the present study. The section must also be improved by adding detailed future research directions.

4.      Lines 57-60 appear to be a repetition of what is included in the following headings. Lines 145-150 appear to have the same issue.

Thank you for your time and consideration.

Round 2

Reviewer 1 Report

Comments and Suggestions for Authors

The authors can answer why the noise can be neglected. In addition, the authors clearly explain why such results come out. Thus, I can now recommend the revised manuscript can be acceptable as it is.

Reviewer 3 Report

Comments and Suggestions for Authors

Dear Doctors, thank you for your time and consideration. 

With best regards, 

The Reviewer